# Crystal structure of a subtilisin-like autotransporter passenger domain reveals insights into its cytotoxic function

Lilian Hor [1], Akila Pilapitiya[1], James A. McKenna [1], Santosh Panjikar [2,3], Marilyn A. Anderson [1], Mickaël Desvaux[4], Jason J. Paxman [1] ✉ & Begoña Heras [1] ✉

Autotransporters (ATs) are a large family of bacterial secreted and outer membrane proteins that encompass a wide range of enzymatic activities frequently associated with pathogenic phenotypes. We present the structural and functional characterisation of a subtilase autotransporter, Ssp, from the opportunistic pathogen *Serratia marcescens*. Although the structures of subtilases have been well documented, this subtilisin-like protein is associated with a 248 residue β-helix and itself includes three finger-like protrusions around its active site involved in substrate interactions. We further reveal that the activity of the subtilase AT is required for entry into epithelial cells as well as causing cellular toxicity. The Ssp structure not only provides details about the subtilase ATs, but also reveals a common framework and function to more distantly related ATs. As such these findings also represent a significant step forward toward understanding the molecular mechanisms underlying the functional divergence in the large AT superfamily.

Subtilases are a diverse group of mostly extracellular proteases found in bacteria, archaea and eukaryotes. The subtilisin-like group of proteins constitute the second largest family of serine proteases after the (chymo)trypsin family of proteases[1]. Subtilases play a role in a wide range of biological functions including bacterial and viral infections, tumorigenesis and metastasis, disease pathogenesis, plant growth and development, and are also utilised in industry worldwide as additives in laundry and dishwashing detergents[2–6]. Although subtilases have been extensively investigated, one unique and largely uncharacterised group of subtilases are found as part of the bacterial autotransporter (AT) protein superfamily.

Autotransporters (ATs) are the largest family of secreted proteins in Gram-negative bacteria with many having important roles in bacterial infection and disease, including adhering to and invading host cells, biofilm formation along with being potent cytotoxins and immunomodulators[6,7]. ATs have a conserved domain structure which includes a signal peptide for Sec-dependent transport across the inner bacterial membrane and a C-terminal translocator domain for secretion of a passenger domain across the outer membrane (Fig. 1a). The surface transported passenger is responsible for the different roles in bacterial pathogenesis (Fig. 1a), whereby specific domain architectures allow ATs to possess distinct functions including enzymatic activities such as protease, lipase and phosphatase activity.

Subtilase ATs comprise one of the two AT families which possess protease activity, with the other being the (chymo)trypsin-like protease ATs which include SPATEs (serine protease autotransporters of Enterobactericeae). The latter group are arguably the best characterised ATs, which integrate a three-stranded β-helix with an N-terminal protease subdomain as part of their passenger[8]. For instance, SPATEs such as the plasmid encoded toxin (Pet), contain a ~600 residue β-helix with an N-terminal (chymo)trypsin-like subdomain which cleave substrates including spectrin leading to

[1]Department of Biochemistry and Chemistry, La Trobe Institute for Molecular Science, La Trobe University, Kingsbury Drive, Bundoora, VIC 3086, Australia. [2]Australian Synchrotron, ANSTO, Clayton, VIC 3168, Australia. [3]Department of Biochemistry and Molecular Biology, Monash University, Clayton, VIC 3800, Australia. [4]INRAE, Université Clermont Auvergne, UMR454 MEDiS, 63000 Clermont-Ferrand, France. ✉e-mail: j.paxman@latrobe.edu.au; b.heras@latrobe.edu.au

cytoskeletal disruption[9]. By comparison, there is less information on subtilase ATs, a large group of proteins found in bacterial pathogens ranging from *Pseudomonas* spp[10–14], *Neisseria meningitidis*[15], and *Bordetella pertussis*[16] amongst others. Some of the better studied subtilase ATs include *B. pertussis* SphB1 and *N. meningitidis* NalP, both lipidated[17,18] and shown to cleave different surface proteins such as filamentous hemagglutinin[16] and the protease IgA or lactoferrin binding protein LbpB, respectively[15,19]. Overall, research to date shows that many subtilase ATs appear to have roles in bacterial pathogenesis such as eliciting cytopathic effects on host cells, conferring serum resistance and biofilm formation, roles which appear to also be common to the (chymo)trypsin-like protease ATs[10,13,20–23].

The first characterised subtilase AT was Ssp (PrtS) from *Serratia marcescens* IFO-3046. *S. marcescens* is an opportunistic pathogen that can infect a range of animals and plants[24,25], and in humans *S. marcescens* is known to causes various infections including respiratory diseases, bloodstream and urinary tract infections, and alarmingly has developed resistance to many commonly used antibiotics[24,26]. The role of Ssp in these infections is unknown; however, a variant of Ssp from *Serratia* sp. A88copa13 (98% identity in passenger) isolated from pine trees was found to be principally responsible for cytotoxicity towards the nematode *Bursaphelenchus xylophilus*, the causative agent of pine wilt disease[27]. Previous studies on Ssp have largely focused on its mechanism of transport from the inner cytosolic membrane to the outer bacterial membrane and its release from the cell surface[28–32]. Like other subtilase ATs, Ssp is thought to remain transiently attached to the bacterial surface, before being cleaved and released into the environment[28]. However, similar to other subtilase ATs, there is no structural and little functional data which exists for Ssp[33].

Here in our investigation of Ssp, we describe the detailed structure/function analysis of a subtilase AT. We reveal that Ssp folds into a unique subtilase domain followed by a short β-helix, an overall layout resembling that of SPATEs. Further, we show that both types of protease ATs also share functional similarities, where we reveal that Ssp can enter epithelial cells and cause cytopathic effects dependent on its subtilase activity. These findings not only serve to bridge the gap between these two large groups of ATs, but also add to the diversity of subtilases and subtilisin-like proteases.

## Results

### The subtilase Ssp is cleaved upon translocation and is proteolytically active

To begin to understand the molecular details of the mostly unexplored AT subtilase sub-group, the gene encoding for full length Ssp from *S. marcescens* IFO-3046[28], which included the N-terminal signal sequence, followed by the passenger and translocator domains, was subcloned into pBAD/*Myc*-His-B and expressed in *E. coli* BL21(DE3). An overexpressed 66 kDa secreted protein which corresponded to the mature form of the cleaved Ssp passenger, was isolated and purified from the culture supernatant for further characterisation. Tryptic digest LC-MS/MS analysis confirmed the Ssp passenger (75% sequence coverage) with data consistent with cleavage at Ala27 and Asp645 releasing the passenger from the translocator[31] (Fig. 1a). This autoproteolysis has been previously noted for Ssp[30], with the mechanism well characterised for other serine protease autotransporters[34]. Recombinant Ssp was also found to be proteolytically active using a fluorogenic casein-based substrate (see below Fig. 3c).

### The Ssp structure reveals a distinct AT three-dimensional architecture

To determine the three-dimensional structure of Ssp by X-ray crystallography, isomorphous crystals were obtained for both unlabelled and selenomethionione (SeMet) labelled Ssp passenger. A preliminary partial model of Ssp was built based on the IcsA autotransporter (PDB: 5KE1)[35] using the homology model server SWISS-MODEL[36].

A number of SeMet datasets were merged, which were used along with the Ssp model to solve the structure using single isomorphous replacement with anomalous scattering (SIRAS) and molecular replacement with single wavelength anomalous diffraction (MRSAD). One molecule of Ssp is present in the asymmetric unit, where the resulting structure was refined against a higher resolution native (2.0 Å) dataset with a final and $R_{factor}/R_{free}$ of 16.3/20.1 (crystallographic statistics in Table 1).

This structure of Ssp revealed that it is composed of an N-terminal subtilisin-like protease domain and a C-terminal three-stranded β-helical domain (Fig. 1b). The subtilase domain of Ssp is positioned directly on top of the β-helical domain, essentially capping the β-helical scaffold (Fig. 1b). Both the subtilase domain and the architectural arrangement of this domain with the β-helical domain has not been observed in AT structures determined to date, and differs from the SPATEs which have their subdomains, including the serine protease domain, protruding from the side of the β-helical stalk (Supplementary Fig. 1)[37–41]. Nonetheless, the presence of the β-helix in Ssp confirms the relationship of subtilase ATs to other important AT groups such as the SPATEs and the self-associating ATs.

### Ssp's subtilase domain has a unique substrate binding cleft

The core architecture of the protease domain of Ssp is typical of the subtilisin-like folds observed in other subtilase enzymes[42]. It is comprised of a globular α/β fold centrally featuring a twisted seven-stranded parallel β-sheet adjacent to two central α-helices and encompassing the active site residues D76, H112 and S341 (Figs. 1c, d and 2c). A disulfide bond (Cys288 to Cys295) connects the last β-strand of the β-sheet to its preceding loop (Fig. 1d). DALI analysis[43] revealed that the most structurally similar protein to Ssp is subtilisin BPN′ from *Bacillus amyloliquefaciens* (31% identity, PDB: 1LW6, Z-score: 32.3) with a r.m.s.d of 2.0 Å over 255 Cα atoms. Superimposition of the two structures confirms a similar central α/β core with the active site residues of Ssp overlaying well with those of subtilisin BPN′ (Fig. 2a, b).

We constructed a site-directed mutant of Ssp, Ssp-S341A, which confirmed the previously identified active site serine of Ssp (29), as it showed no activity against a casein-based substrate (Fig. 3c). In addition, the inability of Ssp-S341A to be released into the culture media when expressed in *E. coli* BL21(DE3), unlike that of wildtype Ssp, verified the role of the Ssp subtilase domain in mediating its self-cleavage and release from the bacterial cell surface. In contrast, when Ssp-S341A was expressed in *E. coli* Top10, it was released into the culture media through cleavage by OmpT[31]. The remaining active site residues Asp76 and His112 were identified from sequence conservation with other subtilases (Supplementary Fig. 2) and their close hydrogen bond distance proximity within the structure.

In comparison to the subtilisin BPN′, the catalytic triad sits at the bottom of a deep cleft formed by two protruding β-hairpins (E1 and E3) on one side and an extended loop (E2) on the other (Fig. 1b, c) which are absent from subtilisin BPN′ (Fig. 2a). This distinctive entrance to the active site, located at the top of the AT, is formed by three major insertions in the Ssp sequence (Supplementary Fig. 2a). Furthermore, the extended loop protrusion (E2) connects to an α-helix that wraps around the side of the subtilase domain of Ssp, another unique addition (Fig. 1c). We assessed the role of the unique finger-like protrusions E1–E3 associated with the active site of Ssp's subtilase domain by generating deletion mutants Ssp-ΔE2, Ssp-ΔE3 and Ssp-ΔE2/E3 (Fig. 3 and Supplementary Table 1). Ssp-ΔE1 was found to be unstable being degraded shortly after secretion. Removal of loop E2 was found to almost completely abolish subtilase activity, whereas deletion of α-helix embellishing E3 β-hairpin had a lesser decrease in activity relative to wildtype Ssp (Fig. 3c). This data supports a role for the protruding loops in substrate recognition and binding for Ssp subtilase function.

Further detailed comparison of the substrate binding site of Ssp and subtilisin BPN′ revealed a notable difference between the two

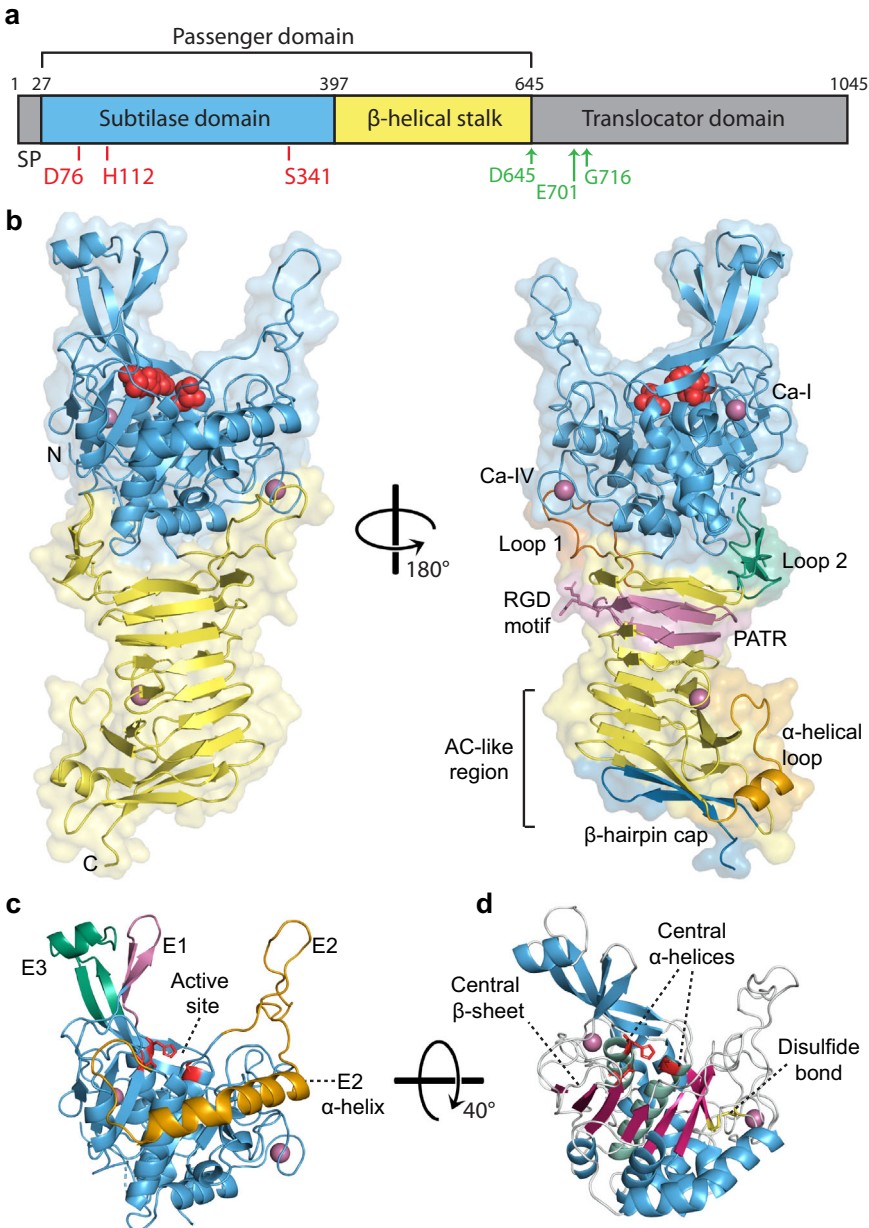

**Fig. 1 | Structure of Ssp. a** Linear schematic of Ssp's primary sequence encompassing an N-terminal signal peptide (SP) and a C-terminal translocator domain flanking the central passenger. Active site residues in the subtilase domain are shown by red line, self-cleavage sites shown by green arrows. **b** Crystal structure showing overall architecture of Ssp passenger in cartoon representation. Subtilase (protease) domain is depicted in blue with active site residues shown as red spheres. The β-helical stalk domain is represented in yellow, with protruding β-helix Loop 1 and Loop 2, coloured in rust and green, respectively. The passenger-associated-transport-repeat (PATR) is displayed in pink and the RGD motif shown as pink sticks, the α-helical loop in orange and the β-hairpin cap at the base of the passenger are coloured in dark blue. Calcium ions are shown as pink spheres with those bound to the protease domain labelled using Dohnalek et al. nomenclature[57]. **c** Protease domain of Ssp in cartoon representation showing the unique active site protrusions (E1–E3). Namely, short β-hairpin extension (E1, pink) long β-hairpin extension (E3, green), and extended loop extension with connected α-helix (E2, orange). Active site is displayed as red sticks. **d** Ssp protease domain with central β-sheet and α-helices shown in hot pink and jade, respectively. Disulfide bond is shown as yellow sticks.

enzymes. Ssp harbours a deletion, equivalent to residues 98–109 of subtilisin BPN' (Fig. 2a, d, and Supplementary Fig. 2a, c), adjacent to the subtilisin BPN' S4 subsite. This results in a truncated α-helix and makes the binding cleft wider (Fig. 2f, g), which conceivably allows for a more extensive substrate binding interface.

The potential individual substrate-binding pockets of Ssp, were identified by overlaying the structure of subtilisin BPN' in complex with the inhibitor CI2 (PDB: 1LW6)[44] with the subtilase domain of Ssp. The bound reactive site loop of CI2 provided a guide to the subsite positions in Ssp (Fig. 2e, g). From this analysis it appears that the Ssp substrate binding site is capable of binding at least five residues P1–P5

(Fig. 2g) on the N-terminal, non-prime side of the scissile bond compared to four residues (P1–P4) for subtilisin BPN' (Fig. 2e). Additionally, the electrostatic properties of the Ssp substrate binding region revealed that it is more positively charged than that of subtilisin BPN' (Fig. 2e, g). Indeed, the afore identified putative S1 and S5 subsites of Ssp appear to be highly positively charged suggesting that it is likely to favour binding of negatively charged amino acids in these positions. Ssp is known to cleave itself at three different positions which releases the passenger from the translocator on the bacterial cell surface (Fig. 1a)[29,31]. Notably, these identified sequences recognised by Ssp (Supplementary Table 2), indeed show a preference for negatively

**Table 1 | Data collection and refinement statistics**

| Data Set | Native Ssp | SeMet Ssp | | |
|---|---|---|---|---|
| | Native-A | SeMet-B | SeMet-C | Merged-D |
| Space group | P1 | P1 | P1 | P1 |
| Unit cell: $a,b,c$ (Å) | 47.48, 55.36, 61.88 | 47.30, 55.27, 61.60 | 47.33, 55.31, 61.61 | 47.31, 55.30, 61.61 |
| $\alpha,\beta,\gamma$ (°) | 91.52, 93.04, 102.76 | 91.54, 92.88, 102.72 | 91.53, 92.88, 102.72 | 91.53, 92.88, 102.72 |
| Total (processed) frames | 3600 (3600) | 3600 (2300) | 3600 (2300) | 7200 (4600) |
| Wavelength (Å) | 0.9537 | 0.9642 | 0.9642 | 0.9642 |
| Oscillation (°) | 0.1 | 0.1 | 0.1 | 0.1 |
| Resolution (Å) | 46.2–2.1 (2.1–2.0) | 20–1.93 | 20–1.93 | 20–1.93 |
| Rmerge (%) | 7.8 (59.7) | 7.6 (28.8) | 7.5 (29.5) | 6.9 (33.0) |
| Rmeas (%) | 9.3 (72.4) | 10.8 (40.7) | 10.6 (41.8) | 8.89 (42.7) |
| $I/\sigma(I)$ | 8.3 (2.0) | 5.59 (1.51) | 5.52 (1.52) | 8.84 (2.01) |
| Completeness (%) | 97.1 (93.3) | 88.8 (85.1) | 89.2 (87.2) | 93.2 (89.2) |
| Multiplicity | 3.3 (3.1) | 1.24 (1.24) | 7.77 (7.70) | 2.37 (2.31) |
| $CC_{1/2}$ (%) | 99.7 (77.4) | 99.3 (85.0) | 99.6 (87.0) | 99.6 (86.0) |
| CCanom | – | (45)10(–3) | (45)13 (–2) | (86)32 (16) |
| Total reflections | 133482 (17143) | 100843 (15826) | 102203 (16544) | 202480 (14068) |
| Unique reflections | 39914 (5596) | 81320 (12706) | 82181 (13132) | 85176 (6064) |
| Refinement | | | | |
| $R_{work}/R_{free}$ | 16.3/20.1 | | | |
| Protein/water atoms | 4611/311 | | | |
| Bonds (Å) | 0.007 | | | |
| Bond angles (°) | 1.006 | | | |
| Ramachandran plot favoured (%) | 97.06 | | | |
| Ramachandran plot allowed (%) | 2.94 | | | |
| B-factors (Å²) Protein | 36.78 | | | |

Values shown in brackets from highest resolution shell.

charged residues in the P1 and P5 position of the substrate, which correlates with the positively charged putative S1 and S5 subsites.

### Ssp subtilase has a short β-helix

The protease domain of Ssp sits on top of the β-helical stalk domain (Fig. 1b). This β-helix is a common scaffold found in the majority of AT structures determined to date[39,45–51]. DALI analysis[43] of solely the β-helical stalk of Ssp indicated that this domain shares low structural similarity to known structures with the closest match being the C-terminal fragment of the AT IcsA from *Shigella flexneri* (15% identity, PDB: 5KE1, Z-score: 15.7) with an r.m.s.d. of 3.0 Å over 197 Cα atoms.

Interestingly, the Ssp β-helix is the shortest seen to date for an AT, comprising of seven right-handed triangular turns (~40 Å in length), with the last three β-strand rungs resembling an autochaperone domain (Supplementary Fig. 3 and Supplementary Table 3), which previous studies have shown are involved in AT passenger folding[52,53]. Interestingly, a short α-helix at turn 7 marks the boundary between the β-helix from the autochaperone region (Fig. 1b).

This size difference of the β-helix is significant when compared to other ATs of known structure which have β-helices between 13–24 turns (87–120 Å in length) (Supplementary Fig. 1). Ssp's β-helix is stabilised by a network of hydrogen bonds formed between the parallel β-strands which form the three-stranded structure and is capped by a C-terminal β-hairpin motif.

There are two loops which extend from the first turn of the β-helix: Loop 1 (residues 405–421) and Loop 2 (residues 431–450) (Fig. 1b), with the latter encompassing two anti-parallel β-strands and a disulfide bond (Cys445 to Cys450). These loops are reminiscent of those present on SPATEs such as Pet subdomains d2 and d4, with d2 found to be involved in binding and internalisation into epithelial cells[54].

The β-helical stalk of Ssp includes a passenger-associated-transport-repeat (PATR) motif (residues 464–496, Fig. 1b). This conserved repeat is found in many ATs and has been demonstrated to be important in the efficient secretion of the passenger[55]. In addition, an RGD motif (residues 482–484) is embedded in the PATR region (Fig. 1b). The RGD motif is a known integrin binding sequence facilitating cell adhesion[56], therefore, may be involved in the biological function of Ssp.

### Ssp is stabilised by calcium

The structural characterisation of Ssp revealed the presence of three calcium binding sites, two in the subtilase domain and one in the β-helical stalk domain (Fig. 1b). The first calcium ion maps to a common high affinity calcium binding site (Ca-I, Dohnalek et al. nomenclature[57]), found in many subtilases such as subtilisin BPN'. In Ssp, Ca-I is coordinated by Glu45, Glu84 Asp125, and has backbone contacts to Lys123, Ala127 and Met129 (Supplementary Fig. 4a). By comparison, the second calcium binding site (Ca-IV, Dohnalek et al. nomenclature[57]) appears to be less common with the site first characterised in sedolisin (formally known as PSCP), a subtilisin-like protease from the structurally related S53 protease family[58]. Structural alignment of Ssp and sedolisin (PDB: 1GA1) shows that Ca-IV is coordinated by similar residues, Asp328, Asp348, and has backbone contacts to Val329, Gly344 and Gly346 in sedolisin, compared to Asp376, Asp383, and has backbone contacts to Val376, Leu377 and Gly381 in Ssp (Supplementary Fig. 4b). The calcium ion found in the β-helical stalk domain is coordinated by the backbone of Pro529 and Gly532 from a loop extending from turn 5, and Glu553 and Asp561 from turn 6 (Supplementary Fig. 4c). Together these turns along with a loop from turn 7 form a pocket for the calcium at the C-terminus of the β-helix.

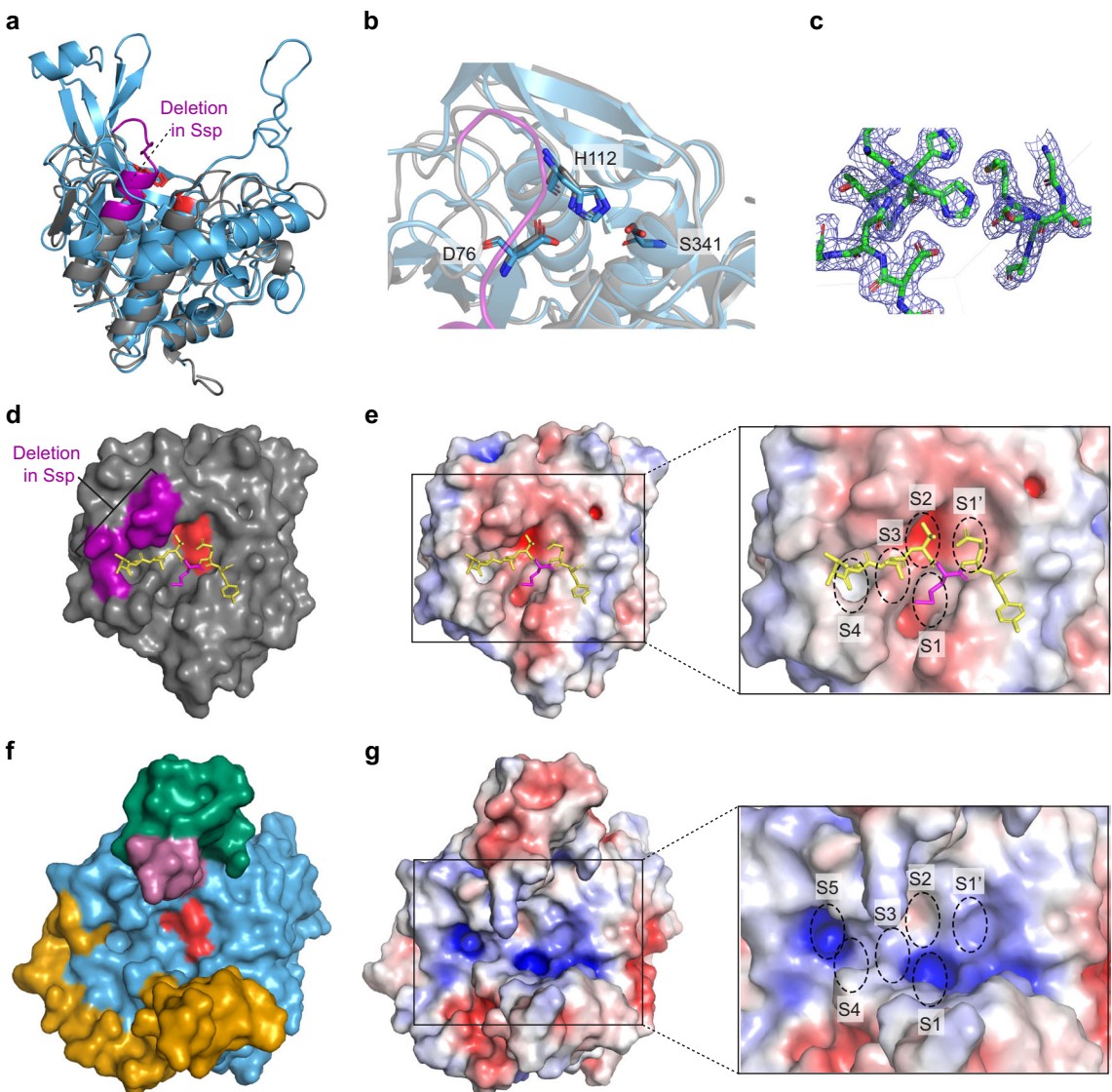

**Fig. 2 | Comparison of the protease domain of Ssp and subtilisin BPN'. a** Overlay of the crystal structure of the subtilase domain of Ssp (blue) and subtilisin BPN' (grey, PDB: 1LW6[44]. Active site residues are shown in red sticks. Subtilisin BPN' residues 98–109 which are missing in Ssp and make the substrate binding cleft wider are shown in purple. **b** Close up of active site residues with Ssp in blue and subtilisin BPN' in grey. Ssp catalytic triad is labelled. **c** 2Fo−Fc electron density map contoured at 1σ encompassing the Ssp subtilase active site. **d** Top-down view of the active site of subtilisin BPN' in complex with inhibitor CI2 (P5-P2' residues displayed for clarity). Subtilisin BPN' residues 98–109 are shown in purple. **e** Electrostatic surface of subtilisin BPN' complexed with CI2 with protease. Subsites are labelled in the inset. **f** Top-down view of the active site of Ssp. Ssp's unique active site extensions are highlighted, namely short β-hairpin extension (E1, pink) long β-hairpin extension (E3, green), and extended loop extension with connected α-helix (E2, orange). Active site is shown in red. **g** Electrostatic surface of Ssp protease domain. Putative protease subsites indicated in the inset. The electrostatic surface potentials were calculated with the APBS plugin in Pymol with electrostatic potential coloured from negative (red) to positive (blue) with a range of ±5 kT/e.

Amongst other roles, calcium is well known to stabilise the structures of many enzymes[42]. Site-directed mutagenesis on the calcium binding site of sedolisin was found to reduce its protease activity[59]. To examine the contribution of $Ca^{2+}$ to the structural stabilisation of Ssp, the apparent thermal melting temperature ($T_m^{app}$) of Ssp was determined. It was found that Ssp is more stable in the presence of $Ca^{2+}$ and is less thermally stable in the presence of the $Ca^{2+}$ chelator EDTA (Supplementary Table 4), with a difference of 4.8 °C. Interestingly, the presence or absence of $Ca^{2+}$ binding had no effect on the subtilase activity of Ssp (Supplementary Fig. 5).

### Ssp causes cell rounding and is internalised by HEp-2 cells

Various protease ATs, from both the SPATE and subtilase families, have been shown to cause mammalian cell rounding, detachment and lysis[10,20–23,60,61], which can result in tissue destruction in vivo. HEp-2 cells

(HeLa derivative) are one of the more common cell lines used in these studies[9,18,55,56]. To further investigate this function, Top10 supernatants containing Ssp were incubated with HEp-2 monolayers, which caused the cells to become rounded and eventually lead to cell detachment (Fig. 4a). This phenotype was not observed in the active site mutant, Ssp-S341A. Furthermore, the Ssp active site protrusion mutants displayed a similar subtilase activity profile against HEp-2 cells as was observed against the casein-based substrate (Fig. 3c) with Ssp-ΔE2 and Ssp-ΔE2/E3, which had ~5% activity against the casein substrate, causing less cell rounding than Ssp-ΔE3, which retained ~80% of the protease activity (Fig. 4a). Collectively, these results demonstrate that Ssp subtilase activity causes cell rounding and detachment, with the finger-like protrusions E2 and E3 playing a role in recognition and binding to cellular targets. Interestingly, although cell rounding and detachment of HEp-2 cells were observed, Ssp did not cause cell lysis as determined

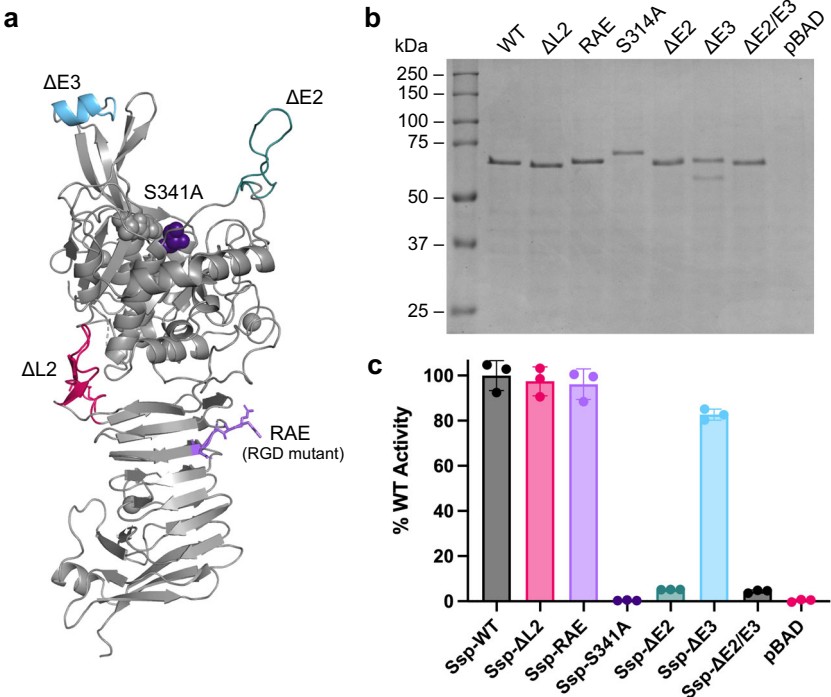

**Fig. 3 | Expression and protease activity of Ssp variants. a** Crystal structure of Ssp in cartoon representation displaying location of mutations. **b** SDS-PAGE analysis of culture supernatant from expression of Ssp variants in *E. coli* Top10. Data is representative of three independent experiments. **c** Protease activity of Ssp variants using a fluorescent casein substrate. Error bars represent standard deviation. Ssp variants include wildtype (WT), deletions of Loop 2 (ΔL2), deletion of active site protrusions E2 and E3 (ΔE2, ΔE3, ΔE2/ΔE3), and site-directed mutants of the active site Ser (S314A) and RGD motif (RAE). pBAD denotes vector only control. Mean is plotted with error bars representing the standard deviation of technical replicates (*n* = 3). Data is representative of three independent experiments.

by LDH release even when the protein was incubated with HEp-2 cells for 24 h (Fig. 4b).

As a number of ATs, such as the SPATE Pet and subtilase AT NalP from *N. meningitidis*, are known to enter mammalian cells to exert their effects[21,62], we sought to determine if Ssp is also internalised by HEp-2 cells. Top10 supernatants containing Ssp were incubated with HEp-2 cells, washed, permeabilised, immunostained with an anti-Ssp polyclonal antibody and observed by confocal microscopy. Imaging showed Ssp to be found in the cytoplasm of the cells which indicates that Ssp is internalised by HEp-2 cells (Fig. 5).

On the Pet β-helix a 57-residue loop (subdomain d2) was found to promote binding and internalisation of Pet into HEp-2 and HT-29 cells[54]. Therefore, we generated deletion mutants of Ssp lacking either β-helix Loop 1 (residues 405−421, Ssp-ΔL1) and Loop 2 (residues 431−450, Ssp-ΔL2) along with generating amino acid substitutions to the RGD cell binding motif (residues 482−484, Ssp-RAE) that is part of the PATR region (Fig. 1a). Both Ssp-ΔL2 and Ssp-RAE were expressed and found to be active at a comparable level to wildtype Ssp (Fig. 3b, c), however, neither Ssp mutant were reduced in the capacity to promote HEp-2 cell rounding, detachment, and internalisation (Fig. 5). Thus, unlike the d2 subdomain in Pet, these β-helix associated features of Ssp are not required for these processes. As Ssp-ΔL1 was not stable after expression, the effect of Loop 1 on Ssp mediated cell rounding and internalisation could not be determined.

Next, we assessed the effect of subtilase activity on internalisation by HEp-2 cells. The active site mutant, Ssp-S341A, showed no entry into the epithelial cells, which suggests that Ssp internalisation is also dependent on its subtilase activity (Fig. 5). This finding is contrary to most previously studied ATs and therefore suggests a different mode of entry into cells by Ssp, where its subtilase function plays an active role. Ssp active site protrusion mutants, Ssp-ΔE2, Ssp-ΔE3 and Ssp-ΔE2/E3, which have attenuated protease activity were also shown to be internalised by HEp-2 cells (Fig. 5), suggesting that even 5% of wildtype activity is sufficient for Ssp internalisation.

## Ssp is toxic in vivo

Given that Ssp induces cytopathic effects on human epithelial cells, we investigated its effects in vivo. *S. marcescens* as an opportunistic pathogen has a diverse host range including insects; thus, the in vivo toxicity of Ssp was tested in a *Galleria mellonella* larvae model. Purified Ssp-WT of varying concentrations was administered by intra-haemocoel injection and larvae mortality was monitored (Table 2). At a dosing above 18.75 nmol/kg (2.5 mg/kg) mortality was observed with 100% death of larvae at the higher doses. This toxicity was found to be dependent on the Ssp subtilase activity, with PMSF inactivated Ssp administered at 150 nmol/kg (10 mg/kg) causing no mortality. Significantly, when purified Ssp-ΔE2, which has ~5% activity compared to wildtype, was administered, no mortality was observed at all doses confirming the importance of the unique finger-like active site protrusions in the activity and thus toxicity of Ssp.

## Discussion

Here, we present the structural and functional characterisation of a subtilase AT, Ssp from *S. marcescens*, which represents a large group of important subtilase ATs within the broader family. Within the extended AT family, Ssp is only the third enzyme class with a structure determined, showing significant differences to the previously determined (chymo)trypsin-like protease and esterase ATs[40,63]. Apart from variation relating to the different enzymatic domains, Ssp was found to incorporate a β-helix, which differs from the esterase ATs, which do not possess a β-helical scaffold as observed in EstA from *Pseudomonas aeruginosa*[63]. The Ssp β-helix is the shortest determined to date for an AT with just seven turns. By comparison, the (chymo)trypsin-like protease ATs have a β-helix over three times longer with 23−24 turns

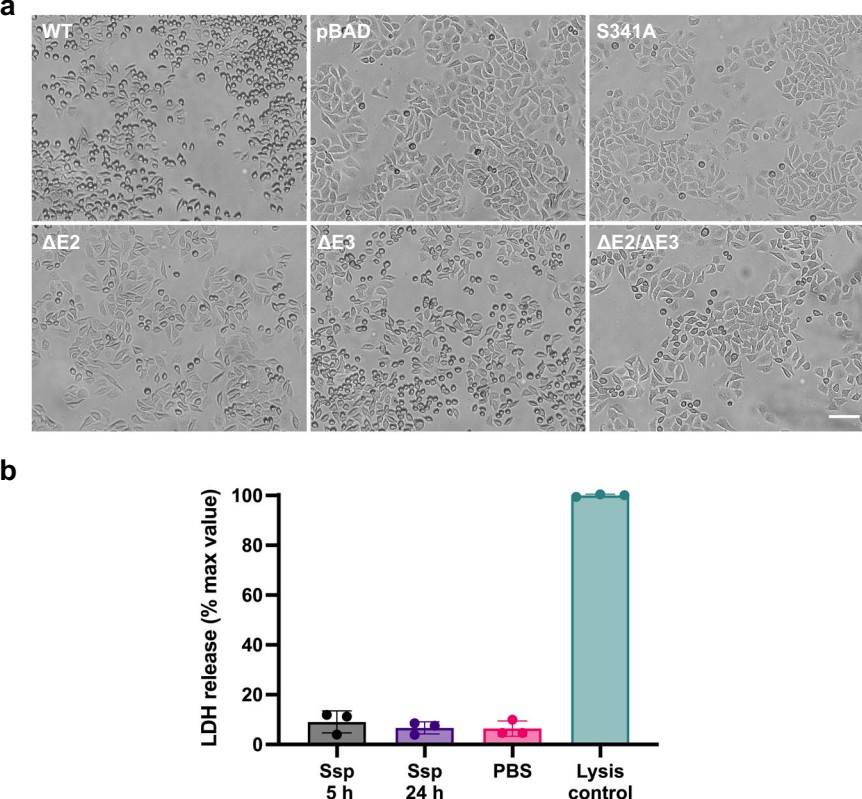

**Fig. 4 | Effect of Ssp on HEp-2 cells. a** HEp-2 cells were incubated with concentrated culture supernatants of Ssp variants (25 μg/mL) for 30 min and imaged by DIC microscopy. Scale bar represents 100 μm. Ssp variants include wildtype (WT), deletion of active site protrusions E2 and E3 (ΔE2, ΔE3, ΔE2/ΔE3), and site-directed mutant of the active site (S314A). pBAD denotes vector only control. Cell rounding was observed WT ~75%, pBAD ~5%, S341A ~5%, ΔE2 ~20%, ΔE3 ~50%, ΔE2/ΔE3 ~10%. **b** Cytotoxicity of Ssp was measured by incubation with HEp-2 cells and the release of LDH measured. Mean is plotted with error bars representing the standard deviation of technical replicates ($n = 3$). Data is representative of three independent experiments.

(Supplementary Fig. 1). This finding is in contrast to previous predictions of a β-helix being absent in subtilase ATs[33].

The Ssp subtilase domain reveals unique features not observed in any of the over 60 subtilases with structures determined to date. In addition to being attached to a 248 residue β-helix, the most notable features of the Ssp subtilase domain includes three finger-like protrusions, two of which form β-hairpins, that surround the active site (Fig. 1b). Extensions around the active site of subtilases are not common occurring in ~13% of subtilases with known structures. Even more unusual is that two of these protrusions, the E2 loop and E3 β-hairpin are unique to Ssp. We found that Ssp's E2 loop and E3 β-hairpin were involved in substrate recognition/binding. This is akin to the disulfide-tethered protrusion near the active site of AprV2 from *Dichelobacter nodosus* (PDB: 3LPC), which has been shown through mutagenesis studies to facilitate substrate-enzyme interactions[3]. These protrusions are in stark contrast to other features associated with subtilase active sites such as the much larger protease-associated subdomains and fibronectin subdomains found in *Streptococcus pyogenes* ScpA (PDB: 3EIF)[64] and tomato subtilase 3 (PDB: 3I6S)[65]. Interestingly, an alignment with other subtilase ATs (Supplementary Fig. 6) has shown that insertions, like the unique finger-like extensions seen in Ssp, are prevalent suggesting that these additions may be a feature adopted by subtilase ATs to increase substrate specificity and/or recognition. Another notable feature of the Ssp subtilase domain is a wide substrate binding cleft, with a preference for binding substrates with negatively charged substrates at the P1 and P5 positions. These characteristics would conceivably increase Ssp's substate specificity.

Ssp's subtilase activity was found to be critical for entry into human epithelial cells. Indeed, the Ssp-S341A active site mutant had no subtilase activity and was unable to enter HEp-2 cells, while the native as well as other proteolytically active Ssp variants, including Ssp-ΔE2 and Ssp-ΔE2/ΔE3, which retained some protease activity, were internalised under the experimental conditions, albeit the latter protrusion mutants at possibly reduced rates. This dependence on subtilase activity for AT entry into eukaryotic cells contrasts with the subtilase AT NalP from *N. meningitidis;* the only other subtilase AT directly shown to enter cells and which does so independent of its subtilase activity[62]. The other AT group shown to enter eukaryotic cells are the (chymo)trypsin-like serine protease ATs. The most well characterised of these proteins are the toxins Pet from enteroaggregative *E. coli* (EAEC) and EspC from enteropathogenic *E. coli* (EPEC), both of which do not require their serine protease activity to enter epithelial cells. Pet enters epithelia using receptor-mediated endocytosis via cytokeratin-8 while EspC utilises the type III secretion system (T3SS)[66,67]. Clearly, Ssp uses a different strategy to enter eukaryotic cells compared to these other ATs. However, given its co-localisation with the cell nucleus (Fig. 5), Ssp may enter via the endosomal pathway to transit via the Golgi apparatus and endoplasmic reticulum on its way to the cell cytosol. The Ssp route of entry may be similar to that of the recently investigated (chymo)trypsin-like protease ATs TagB, TagC and Sha from extra-intestinal pathogenic *E. coli* (ExPEC), which were shown to require protease activity in order to enter human cells[68].

Ssp subtilase activity was also found to promote cytotoxic activity against cellular targets. This was shown through the addition of Ssp causing cell rounding and detachment of human epithelial cells, which was not observed using the active site mutant and to a lesser extent with the protrusion mutants. Collectively, these results showed that the extent of cell rounding reflects the relative protease activity of the

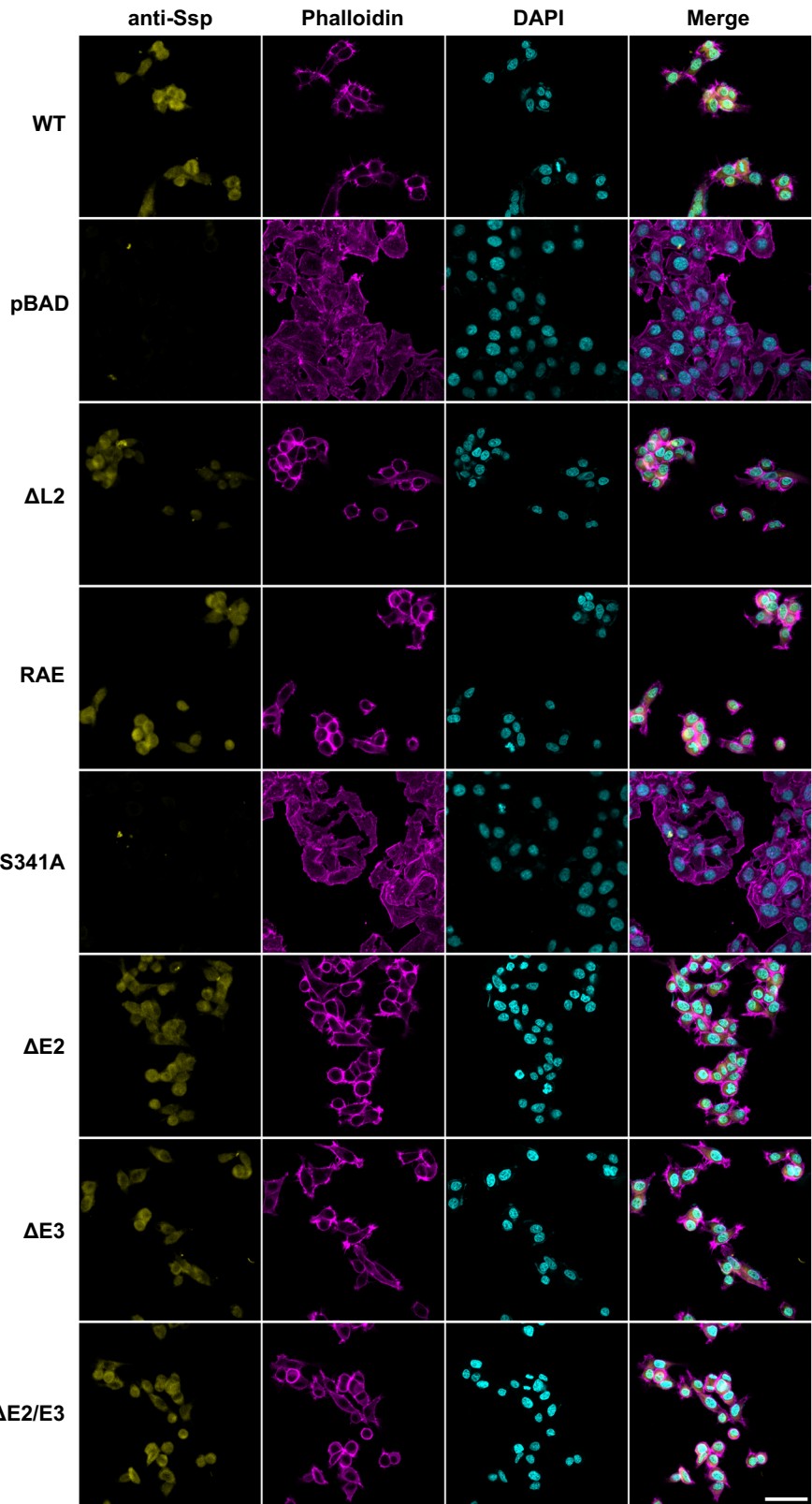

**Fig. 5 | Internalisation of Ssp variants by HEp-2 cells.** Confocal microscopy images of HEp-2 cells incubated with concentrated culture supernatants of Ssp variants (25 µg/mL) for 5 h. Ssp was visualised with anti-Ssp polyclonal antibody followed by Alexa Fluor Plus 647 conjugated secondary antibody (yellow), actin cytoskeleton was stained with phalloidin (magenta) and nucleus stained with DAPI (cyan). Ssp variants include wildtype (WT), deletions of Loop 2 (ΔL2), deletion of active site protrusions E2 and E3 (ΔE2, ΔE3, ΔE2/ΔE3), and site-directed mutants of the active site Ser (S314A) and RGD motif (RAE). pBAD denotes vector only control. Images are representative of cells observed from at least three independent experiments. Scale bar represents 50 µm.

**Table 2 | Ssp toxicity**

| Dose (nmol/kg) | Ssp-WT | Ssp-ΔE2 | Ssp-PMSF |
|---|---|---|---|
| 150 | 0% | 100% | 100% |
| 75 | 0% | 100% | ND |
| 37.5 | 33.3% | 100% | ND |
| 18.75 | 100% | 100% | ND |
| 9.34 | 100% | 100% | ND |
| 0 | 100% | 100% | ND |

*ND* Not determined, *NB* 150 nmol/kg of Ssp-WT is equivalent to 10 mg/kg.
Percentage of surviving *G. mellonella* larvae after injection with purified Ssp variants after 72 h.

added Ssp variant (Figs. 4a and 5). This was further demonstrated in vivo in a *G. mellonella* model, whereby wildtype Ssp caused larvae mortality at doses above 18.75 nmol/kg (2.5 mg/kg) while PMSF inactivated Ssp and active site protrusion mutant Ssp-ΔE2, which has attenuated subtilase activity, showed no mortality at the highest dosing of 150 nmol/kg (10 mg/kg). The relationship between subtilase AT activity and cytotoxicity has previously been demonstrated in PfaI from *P. fluorescens* and Pta from *P. mirabilis* where only proteolytically active AT were found to be cytotoxic to fish and bladder epithelia, respectively[10,20]. In addition, this link has been further elucidated in the (chymo)trypsin-like protease ATs, whereby upon entering epithelia the serine proteases of both Pet and EspC digest fodrin associated with the cell cytoskeleton to cause rounding and detachment of the cells[21,22]. It is indeed plausible that along with promoting cell entry, the Ssp subtilase also digests intracellular targets to promote its cytotoxic effects, but this remains to be confirmed.

The β-helix scaffold is common throughout the ATs has been found to have roles in binding host and bacterial factors regardless of AT group. A loop extension derived from the Pet β-helix termed the d2 subdomain was revealed to interact with the cytokeratin-8 receptor on epithelial cells prior to entry[66]. However, deletion of the equivalent loop on the Ssp β-helix (Ssp-ΔL2) had no observable effect on entry into epithelial cells. By comparison, the β-helix of AT adhesins such as uropathogenic *E. coli* UpaB were found to directly bind cellular surfaces[49]. Inspection of the Ssp β-helix revealed a RGD motif as a possible cell binding site. Unfortunately, disruption of this motif had no noticeable effects on Ssp interaction with epithelial cells. The β-helices of other AT adhesins such as Ag43a were found to be involved in self-association to promote bacterial phenotypes such as biofilm formation[48,50]. However, the Ssp β-helix is unlikely to have this role given its monomeric nature as judged by size-exclusion chromatography and that there is no evidence of Ssp mediated bacterial aggregation (Supplementary Fig. 7). Further, Ssp is known not to remain at the bacterial surface but to be cleaved and released into the environment by the action of its own subtilase along with other surface proteases. In summary, we are yet to define a role for the unusual Ssp β-helix.

Overall, this structure/function analysis of the subtilase Ssp has provided valuable insights into this large group of ATs, and reveals unique structural features not shared with either subtilases or autotransporters that are required for its cytotoxic activity. Nevertheless, there are shared similarities which are also insightful. The fold shared between the Ssp subtilase domain and other subtilases suggests a common origin. Further, the Ssp architecture consisting of an N-terminal protease attached to a C-terminal β-helix reveals a common framework observed in other ATs such as the (chymo)trypsin-like protease ATs. Elucidation of the Ssp structure represents a significant step forward toward understanding the molecular mechanism underlying the functional divergence in the large AT superfamily. Thus, it will be of great importance to uncover other functionally relevant molecular features shared amongst the wider AT and enzyme families, as further ATs are investigated in molecular detail.

## Methods

### DNA constructs and mutagenesis
The gene encoding for Ssp from IFO-3046[28] was codon optimised for expression in *E. coli*, synthesised and subcloned into pBAD/*Myc*-His-B by Invitrogen. Mutations to Ssp-pBAD were carried out using either the Stratagene QuikChange II method with Q5 High-Fidelity DNA polymerase (NEB, M0491S) and DpnI (NEB, R0176S) or the Q5 Site-Directed Mutagenesis Kit (NEB, E0552S). Primers used are detailed in Supplementary Table 5 and list of mutants generated in Supplementary Table 1. Successful mutations were verified by dideoxynucleotide sequencing (Macrogen, Korea).

### Protein expression and purification
*E. coli* BL21(DE3) harbouring the Ssp-pBAD/Myc-His-B plasmid was grown in LB supplemented 100 μg/mL ampicillin at 37 °C. Expression of Ssp was induced by addition of 0.02% L-arabinose at 30 °C for 4 h. The media was harvested, filtered and concentrated ~40 times using Vivaflow 200 (Sartorius, VF20P2) crossflow cassettes at 4 °C before dialysis in 20 mM Tris, pH 9.0. The secreted protein was purified by two round of anion exchange chromatography, HiTrap Q FF (GE Healthcare, 17515601) and Mono Q 10/100 GL (GE Healthcare, 17516701), with 20 mM Tris, pH 9.0 and elution with 1 M NaCl. Recombinant Ssp was further purified by gel-filtration chromatography using a HiLoad Superdex 200 16/600 (GE Healthcare, 28989335) pre-equilibrated with 25 mM HEPES, 150 mM NaCl, pH 7.0.

SeMet labelled Ssp was expressed in minimal media supplemented with 50 μg/mL of selenomethionine inducing with 0.5% L-arabinose at 20 °C overnight[69]. Ssp-SeMet was purified as described above.

Ssp mutants were expressed in *E. coli* Top10 cells in LB inducing with 0.2% L-arabinose at 20 °C overnight. Media was concentrated using Ultra-0.5 Centrifugal Filter Unit (Amicon, UFC5010BK). Protein was quantified by densitometry using SDS-PAGE with purified Ssp as a standard using Image Lab 6.0 (Bio-Rad). Ssp-ΔE2 was purified in the same manner as Ssp-WT.

### Protease activity
Protease activity was measured using EnzChek Protease Assay Kit, green fluorescence (Invitrogen, E6638) in 50 mM Tris, 150 mM NaCl, 0.005% Triton X-100, pH 7.4 using a FLUOstar Omega plate reader (BMG Labtech) at Ex/Em 485/520 nm. Typically, 50 nM Ssp was used in the assay.

### Thermal melts
Thermal denaturation of Ssp was monitored in an Agilent Cary 3500 Multicell UV-Vis spectrophotometer. The thermal melts were conducted from 20–90 °C monitoring at 280 nm using 10 mm pathlength ultra-micro quartz cuvettes (Hellman), with a ramp rate of 0.5 °C/min. Data was collected at a 0.5 mg/mL Ssp in 25 mM HEPES, 150 mM NaCl, pH 7.0 (150 μL) with the addition of either 10 mM EDTA, 10 mM CaCl$_2$ or no additive. Data was fitted to the following equation to estimate the apparent melting temperature ($T_m^{app}$):

$$y = \frac{k}{1+k}\left[(u + u_1x) - (l + l_1x)\right] + l + l_1x \qquad (1)$$

Where $k = e^{\left[\frac{h}{1.987(x+273.15)}\right]\left[\frac{x+273.15}{t+273.15}-1\right]}$, $y$ is absorbance, $x$ is temperature, $h$ is enthalpy, $t$ is $T_m^{app}$, $u$ is folded absorbance, $l$ is unfolded absorbance, and $u_1$ and $l_1$ are linear corrections for folded and unfolded as function of temperature, respectively.

### Crystallisation and diffraction data measurement
Ssp, native and SeMet, were crystallised using the hanging drop vapour diffusion method with drops containing 1 μL of protein solution

(15 mg/mL in 25 mM HEPES, 150 mM NaCl, pH 7.0) and 1 μL of reservoir solution (0.2 M potassium iodide, 20% (w/v) PEG 3350, 0.1 M HEPES, pH 7.0 for native Ssp, and 0.2 M potassium iodide, 23% (w/v) PEG 3350, 0.1 M HEPES, pH 6.8 for Ssp-SeMet) at 20 °C. Crystals appeared within 5 days. For X-ray data collection, the Ssp crystals were soaked in cryoprotectant solution containing reservoir solution made up in glycerol (20% v/v) and directly flash cooled in liquid nitrogen. Both native and anomalous data were collected for 360° using 0.954 Å at 100 K with an EIGER x 16 M detector with 0.1 degree per frame with 0.02 s exposure at the Australian Synchrotron on the MX2 Beamline[70]. The data were indexed and integrated using XDS[71]. The native data were scaled using AIMLESS[72] to 2.0 Å resolution, belonging to space group P1 with cell dimensions of a = 47.48 Å, b = 55.36 Å, c = 61.88 Å and α = 91.52°, β = 93.04° and γ = 102.76°. This was consistent with one molecule in the asymmetric unit. The anomalous data from SeMet crystals were analysed using BLEND[73] for isomorphous unit cells, whereby two datasets were merged using POINTLESS and AIMLESS[72]. See Table 1 for all data-collection and processing statistics.

## Structure determination and refinement

The structure of Ssp was solved using a combination of SIRAS method and MRSAD phasing protocol in the Auto-Rickshaw automated crystal structure determination platform[74]. FA values were calculated using the programme SHELXC[75] with the native and the merged SeMet dataset by combining isomorphous and anomalous signal. A total of nine selenium atoms were found per asymmetric unit using *SHELXD*[76]. After atom refinement and density modification, an initial model was generated using the programme BUCCANEER[77] with SAD refinement, and then further improved using the MRSAD phasing protocol of Auto-Rickshaw[74,78]. The resulting model was refined against the native dataset using REFMAC5[79] and further built using COOT[80]. The quality of the Ssp model was assessed by MolProbity[81]. Ramachandran statistics showed 97% of residues in the most favoured region and 3% in the allowed regions. Refinement values are given in Table 1. Molecular figures were generated using PyMOL[82]. The PDB ID code is 8E7F.

## Cell culture

Bovine serum albumin (BSA) was purchased from Scientifix (BSAS-0.1). HEp-2 was kindly gifted by Dr. Natalie Borg (RMIT, Australia). HEp-2 is a known human HeLa carcinoma contaminant. It is a widely used cell line in autotransporter research and as such, we chose to use the cell line and keep the reference to HEp-2 in the paper for consistency. The cell line was not authenticated. HEp-2 cells were cultured in Dulbecco's modified Eagle's medium (DMEM, Gibco, 11965092) supplemented with 10% foetal bovine serum (FBS, Corning, 35-076-CV) and 2 mM L-Gln (Gibco, 25030081). Cells were cultured at 37 °C in a humidified atmosphere with 5% $CO_2$ and harvested using 0.25% trypsin-EDTA (Gibco, 25200056). All incubations were carried out at 37 °C in a humidified atmosphere with 5% $CO_2$ unless specified. Proteins and concentrated media were sterile filtered with Costar Spin-X centrifuge tube filters (Corning, 8161) before applying to cells.

## Lactate dehydrogenase (LDH) cytotoxicity assay

HEp-2 cells were seeded at a density of 2000 cells/well in 96 well plates (Falcon, 353072) in DMEM, 5% FBS and 2 mM L-Gln. Cells were treated with Ssp for 5 h or 24 h. CyQUANT LDH Cytotoxicity Assay (Invitrogen, C20301) was used to determine LDH activity according to manufacturer's instructions.

## Microscopy

Immunodetection of Ssp was carried out using purified rabbit polyclonal serum raised against purified passenger of Ssp at the Walter and Eliza Hall Antibody Facility (Australia). HEp-2 cells were seeded at a density of $2.0 \times 10^5$ cells/well on 18 mm coverslips (Marienfeld, 0117580) in 12 well plates (Greiner, 665180) for confocal microscopy or

$1.0 \times 10^5$ cells/well in 24 well plates (Greiner, 662160) for DIC microscopy in DMEM, 10% FBS and 2 mM L-Gln. After overnight incubation, the cells were washed with phosphate buffered saline (PBS) and the media changed to RPMI 1640 (Gibco, 11875093).

Confocal microscopy began with 25 μg/mL of concentrated supernatants of Ssp variants added per well and the cells incubated for 5 h before washing with PBS (this concentration is based on the concentrations previously used for the characterisation of cytotoxic autotransporters (30–37 μg/mL)[23,66] as well as the Ssp supernatant concentration previously detected by SDS-PAGE in nematicidal *Serratia* sp.[27]). For slide preparations, all incubations were performed at room temperature and washes were conducted after each step with PBS-T (PBS with 0.05% Tween-20) unless specified. Cells were fixed with 4% formalin/PBS for 10 min before washing with PBS with 100 mM glycine. The fixed cells were permeabilised with 0.2% Triton X-100/PBS for 5 min and blocked with 2% BSA/PBS for 1 h or overnight at 4 °C. The coverslips were then incubated with Ssp polyclonal antisera (1:4) for 1.25 h, Alexa Fluor Plus 647 conjugated goat anti-rabbit secondary antibody (1:200, Invitrogen, A32733) for 1 h, Phalloidin-iFluor 555 Reagent (1:1000, Abcam, ab176756) for 30 min and DAPI (1 μg/mL, Sigma, D9542) for 5 min in the dark. Coverslips were mounted on slides using SlowFade Diamond Antifade Mountant (Invitrogen, S36972) and imaged using Zeiss LSM 780 confocal microscope at 40× magnification. For the anti-Ssp channel, a Z-stack comprising of 10 slices from the top to the bottom of the cells was taken and the Z-stack projected using the sum function in FIJI[83]. Images were processed in FIJI[83] using the BIOP Channel Tools plugin.

Differential interference contrast (DIC) microscopy began with concentrated supernatants of Ssp variants (12.5 μg, 25 μg/mL) added per well and the cells incubated for 30 min before washing with PBS. Cells were imaged using Zeiss Axio Observer microscope at 10× magnification. Images were processed in FIJI[83].

## *Galleria mellonella* toxicity assay

*Galleria mellonella* larva were reared at 19 °C in refrigerated incubator (TRIL-495-1-SD, Thermoline Scientific) on a diet of 46% (w/w) Farex multigrain cereal (6 month+, Heinz), 22% (w/w) glycerol (Chem-Supply), 22% (w/w) honey (Capilano), 7% (w/w) deionized water and 3% (w/w) yeast extract (Oxoid, LP0021B).

Groups of six larvae (average mass of 165 mg) selected at random, were injected with 20 μL of purified Ssp variant diluted in sterile PBS through the last right pro-leg using a 1 mL U-100 insulin syringe (Terumo, 29 G × 13 mm) attached to a NE-1000 syringe pump (New Era). The inoculated larvae were incubated at 37 °C and mortality assessed daily. Larvae were considered dead if no response was observed following physical stimulus with tweezers.

## Aggregation assay

Ssp-pBAD or pBAD vector control in *E. coli* Top10 was cultured in LB with 100 μg/mL ampicillin at 37 °C with shaking. Expression of Ssp was induced by addition of 0.2% L-arabinose at 20 °C overnight. The culture (800 μL) was added to semi-micro cuvettes (Greiner) and the $OD_{600}$ measured for overtime at room temperature (Molecular Devices SpectraMax M5).

## Reporting summary

Further information on research design is available in the Nature Portfolio Reporting Summary linked to this article.

## Data availability

The crystallography, atomic coordinates, and structure factors reported in this paper have been deposited in the Protein Data Bank, under PDB ID 8E7F. The following structural models from the PDB were also used in this study: 1LW6 (subtilisin BPN'), 5KE1 (IcsA autotransporter), 1GA1 (sedolisin), 3LPC (AprV2 from *Dichelobacter*

*nodosus*), 3EIF (*Streptococcus pyogenes* ScpA) and 3I6S (tomato subtilase). Source data are provided with this paper.

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

## Acknowledgements

This research was undertaken in part using the MX2 beamline at the Australian Synchrotron, part of ANSTO, and made use of the Australian Cancer Research Foundation (ACRF) detector. The authors would like to acknowledge Peter Lock and Chad Johnson from the La Trobe University Bioimaging Platform. This work was supported by the Australian Research Council (ARC) project grants (DP150102287, DP180102987, DP210100673), Fellowship (FT130100580) and a National Health and Medical Research Council (NHMRC) project grant (GNT1143638).

## Author contributions

B.H., J.J.P. and M.D. designed the research. L.H., A.P. and S.P. performed the experiments. B.H., J.J.P., L.H. and S.P. analysed the data. B.H., J.J.P., J.A.M. and M.A.D. supervised aspects of the work. All authors contributed to the interpretation of the results. L.H., J.J.P. and B.H. wrote the manuscript. All authors contributed to the critical revision of the manuscript, read, and approved the final manuscript.

## Competing interests

The authors declare no competing interests.
