## [Peer review file · Nature Communications]

REVIEWER COMMENTS

Reviewer #1 (Remarks to the Author):

In this well-written manuscript, Hor and colleagues report the crystal structure of the *Serratia marcescens* Ssp subtilase passenger domain, representing the first crystal structure of a protein in this group of autotransporters and this group of subtilases, which have broad biological functions. The crystal structure reveals a unique binding cleft compared to other subtilisin-like proteins and both shared and unique structural features compared with other autotransporters with protease activity. Based on the crystal structure, several mutants are generated and tested for proteolytic activity, the ability to enter human cells, the ability to cause cell rounding and lifting (cytotoxicity), and virulence in a *Galleria* model. The catalytic site is identified and confirmed experimentally. The experiments are logical and generally well-designed with appropriate controls, and the conclusions appear sound.

Specific comments include the following:

1. The title of the paper would more accurately read "... crystal structure of a subtilisin-like autotransporter passenger domain reveals ...", inserting "passenger domain," since the crystal structure does not represent the full protein.
2. Figs 4+5: The reason for using the HEP-2 cell line is not described. This cell line is well-characterized as a HeLa-contaminated line, a point that should be acknowledged in the text.
3. Fig 5: Based on the small size of the images, it is difficult to see, but it appears that Ssp colocalizes with the nucleus. Is this correct? If so, it requires some discussion. It would be helpful to see an enlarged view of WT Ssp-infected cells to better appreciate the subcellular localization.
4. Table 2: It is unclear why PMSF-treated Ssp was used in the in vivo model rather than the S341A catalytic site mutant, as this mutant was used elsewhere in the study. As a control for PMSF-treated Ssp, the authors should include PMSF only (no Ssp).
5. Can the authors exclude the possibility that the Ssp S341A mutant is non-cytotoxic simply because it cannot enter cells rather than because of a direct on cytotoxic activity?
6. Fig S7: It is not surprising that no aggregation is seen in TOP10 cells expressing Ssp as it can be cleaved from the surface by itself or through action of OmpT. The authors should address this limitation. Expression of Ssp S341A in an ompT mutant *E. coli* background (BL21/DE3) would provide a better condition to test aggregation.
7. Minor points:
 - a. Line 73, typo, delete "of".

- b. Line 74, insert “and” before “Bordetella”
- c. Line 232, typo, delete duplicated “in”.
- d. Line 269, the word “Significantly” seems out of place.
- e. Line 406-407, what is “485-12”? Should it not just be 485?

Reviewer #2 (Remarks to the Author):

The manuscript “The first crystal structure of a subtilisin-like autotransporter reveals insights into its cytotoxic function” by Hor et al. describes a novel crystal structure of an autotransporter passenger and some assays on its putative function.

In general the manuscript is well written and presents the interesting novel structure of a subtilisin-like autotransporter, Ssp. It shows interesting features in both the enzymatic part (the subtilisin subdomain) and the more structural part that is the β -helical stem that forms the base of the structure. The authors also present mutants of the Ssp protein that support their assumptions on the enzymatic activity. They then investigate some functional features of this protein, mostly guided by the work on chymotrypsin autotransporters, and show that the purified protein has cell-rounding activity, notably without lysing the cells. They also show that purified proteins appear to internalize into cultured cells, whereas an active-site mutant does not.

I have a few major comments and some suggestions for improvement:

1: It is strange, but not discussed, that some of the protrusion mutant show hardly cell-rounding, but are internalized when added to cultured cells. In fact, in Fig.5 internalization of all but the active site mutant appears at the same level, which is not expected in view of the difference in protease activity, if that activity is, indeed, needed for internalization, as suggested here. In fact the active site mutant is not detected at all. Is there an explanation for this observation? The cells are permeabilized for antibody binding (should be added to the main text). This could have influenced the uptake, but there is no control for that. Could the washing and permeabilization not have caused the internalization. Finally, is there cell-rounding observed in Fig. 5 for the cells that take up the protein? This requires better explanation.

2: The relatively short β -helical stalk with the subtilisin domain directly on top is a unique AT structure not previously shown. One then wonders how well-studied subtilisin-like ATs (SphB1 and NaIP) or other predicted subtilisin-like ATs would look. Do models of these passengers have the same unique features (protrusions around the active site, Ca^{2+} binding sites, short β -helix) and can this be modelled? This discussion is lacking; the authors choose only to compare the structure to SPATES and the esterase AT, for which structures are available, but that are clearly of a different AT class. However, the comparison to other subtilisins would support the novelty and the understanding of the structural diversity of AT passengers in general.

3. Line 73 and other places; remarks that “little information” is available on subtilisin ATs. In fact the functions of subtilisin-like ATs SphB1 and NaIP have been studied in quite some detail, to an extent similar to most SPATEs. Importantly, both proteolytically process other virulence factors at the bacterial cell surface (filamentous haemagglutinin protein for *Bordetella*, IgA protease for *Neisseria meningitidis*). This is not mentioned anywhere in the manuscript, nor considered as a function for Ssp, whereas it is a striking similarity between subtilisins not shared with the SPATEs.

other comments/suggestions:

4: The authors compare their structure in Fig S1 to the SPATE Pet. However, the cartoon shown appears not the most suitable: 1. the cartoon shown lacks secondary structure detail in its C-terminal part, most notably for the autochaperone domain and also the β -helical rungs that are present in Ssp. Better structures with detail in that region are available 2. the autochaperone domain of Ssp is comparable to that of IcsA, so one would like to see how that compares in this image as well.

5. The remark in line 57 (ATs “are increasingly gathering attention”) is overstating it. ATs are studied over 30 years and there is not a sudden increase in AT papers. This should be re-phrased.

NCOMMS-22-40427 Revision

We thank the editor and all reviewers for their detailed comments. Please see below our response to the suggested revisions.

REVIEWER COMMENTS

Reviewer #1 (Remarks to the Author):

In this well-written manuscript, Hor and colleagues report the crystal structure of the *Serratia marcescens* Ssp subtilase passenger domain, representing the first crystal structure of a protein in this group of autotransporters and this group of subtilases, which have broad biological functions. The crystal structure reveals a unique binding cleft compared to other subtilisin-like proteins and both shared and unique structural features compared with other autotransporters with protease activity. Based on the crystal structure, several mutants are generated and tested for proteolytic activity, the ability to enter human cells, the ability to cause cell rounding and lifting (cytotoxicity), and virulence in a *Galleria* model. The catalytic site is identified and confirmed experimentally. The experiments are logical and generally well-designed with appropriate controls, and the conclusions appear sound.

We thank the reviewer for their comments.

Specific comments include the following:

1. The title of the paper would more accurately read "... crystal structure of a subtilisin-like autotransporter passenger domain reveals ...", inserting "passenger domain," since the crystal structure does not represent the full protein.

Title has been amended accordingly.

2. Figs 4+5: The reason for using the HEP-2 cell line is not described. This cell line is well-characterized as a HeLa-contaminated line, a point that should be acknowledged in the text.

Added line (Page 5 line 240-241):

HEP-2 cells (HeLa derivative) are one of the more common cell lines used in these studies^{9,18,56,57}.

3. Fig 5: Based on the small size of the images, it is difficult to see, but it appears that Ssp colocalizes with the nucleus. Is this correct? If so, it requires some discussion. It would be helpful to see an enlarged view of WT Ssp-infected cells to better appreciate the subcellular localization.

As per the Pet autotransporter and many other toxins that enter cells (PMID: 24327340; 33418946) we think the Ssp co-localises to the Golgi apparatus and endoplasmic reticulum associated with the nucleus, as part of its possible endosomal entry into the cell cytosol.

We have added the following to the discussion (Page 8 line 337-339) :

"However, given its co-localisation with the cell nucleus (Figure 5), Ssp may enter via the endosomal pathway to transit via the Golgi apparatus and endoplasmic reticulum on its way to the cell cytosol."

4. Table 2: It is unclear why PMSF-treated Ssp was used in the in vivo model rather than the S341A catalytic site mutant, as this mutant was used elsewhere in the study. As a control for PMSF-treated Ssp, the authors should include PMSF only (no Ssp).

We thank the reviewer to their attention to detail. At the time we felt PMSF-treated Ssp was the 'ideal control' as we could use an identical batch of purified wildtype protein. We did not include a PMSF only because free PMSF was removed from sample before it was used in the experiments. As the reviewer may know PMSF binds covalently to the active serine. PMSF is quite toxic so the 100% survival of the *G. mellonella* with Ssp-PMSF added further confirms the removal of all excess PMSF.

5. Can the authors exclude the possibility that the Ssp S341A mutant is non-cytotoxic simply because it cannot enter cells rather than because of a direct on cytotoxic activity?

Thank you for the great question. Unlike the Pet autotransporter whereby protease activity is not required for cell entry but it is required for cytotoxicity, these two activities are difficult to distinguish for Ssp. For example, the Ssp- Δ E2 and Ssp- Δ E2/ Δ E3 mutants, which showed reduced subtilase activities, were found to enter HEP-2 cells to a similar extent as for WT under the experimental conditions (Fig 5 anti-Ssp panels), but displayed reduced effects on cell rounding and cytotoxicity (Fig 4 and Fig 5), indicating the relationship between the latter effects and the subtilase activity. Further, the cell rounding and detachment of cells for Ssp WT is reminiscent of the effects observed for the digestion of intracellular targets for the subtilase autotransporters PfaI (PMID: 19447960) and Pta (PMID: 18430084) along with the serine protease autotransporter Pet (PMID: 10225873) and EspC (PMID:15155671), suggesting that the Ssp subtilase also digests intracellular targets. We have further clarified these points in the discussion. Please see also our response to reviewer 2 question 1.

6. Fig S7: It is not surprising that no aggregation is seen in TOP10 cells expressing Ssp as it is can be cleaved from the surface by itself or through action of OmpT. The authors should address this limitation. Expression of Ssp S341A in an ompT mutant E. coli background (BL21/DE3) would provide a better condition to test aggregation.

We thank the reviewer for the comment and good advice. We have now added the following to the discussion (Page 9, lines 371-373):

“Further, Ssp is known not to remain at the bacterial surface but to be cleaved and released into the environment by the action of its own subtilase along with other surface proteases.”

The following has been added to the S7 figure legend:

“Note: Ssp is gradually cleaved and released by the actions of its own subtilase along with OmpT under these conditions.”

7. Minor points:

- a. Line 73, typo, delete “of”. ~~removed~~
- b. Line 74, insert “and” before “Bordetella” ~~added~~
- c. Line 232, typo, delete duplicated “in”. ~~removed~~
- d. Line 269, the word “Significantly” seems out of place. - ~~removed~~
- e. Line 406-407, what is “485-12”? Should it not just be 485? - ~~removed -12~~

Reviewer #2 (Remarks to the Author):

The manuscript “The first crystal structure of a subtilisin-like autotransporter reveals insights into its cytotoxic function” by Hor et al. describes a novel crystal structure of an autotransporter passenger and some assays on its putative function.

In general the manuscript is well written and presents the interesting novel structure of a subtilisin-like autotransporter, Ssp. It shows interesting features in both the enzymatic part (the subtilisin subdomain) and the more structural part that is the β -helical stem that forms the base of the structure. The authors also present mutants of the Ssp protein that support their assumptions on the enzymatic activity. They then investigate some functional features of this protein, mostly guided by the work on chymotrypsin autotransporters, and show that the purified protein has cell-rounding activity, notably without lysing the cells. They also show that purified proteins appear to internalize into cultured cells, whereas an active-site mutant does not.

I have a few major comments and some suggestions for improvement:

1: It is strange, but not discussed, that some of the protrusion mutant show hardly cell-rounding, but are internalized when added to cultured cells. In fact, in Fig.5 internalization of all but the active site mutant appears at the same level, which is not expected in view of the difference in protease activity, if that activity is, indeed, needed for internalization, as suggested here. In fact the active site mutant is not detected at all. Is there an explanation for this observation? The cells are permeabilized for antibody binding (should be added to the main text). This could have influenced the uptake, but there is no control for that. Could the washing and permeabilization not have caused the internalization. Finally, is there cell-rounding observed in Fig. 5 for the cells that take up the protein? This requires better explanation.

We thank this reviewer for their questions.

To address “**The cells are permeabilized for antibody binding (should be added to the main text)**” we have added to the main text (Page 6 lines 256-258):

“Top10 supernatants containing Ssp were incubated with HEp-2 cells, washed, permeabilised, immunostained with an anti-Ssp polyclonal antibody and observed by confocal microscopy.”

In response to your questions, the active site mutant Ssp-S341A had no subtilase activity and did not enter HEp-2 cells, where we suggest from these findings that subtilase activity is required for HEp-2 internalisation. With the knowledge that Ssp-S341A could not be internalised we subsequently used it as a control for HEp-2 Ssp internalisation with regard to washing and permeabilisation.

In terms of internalisation of the protrusion mutants Ssp- Δ E2 and Ssp- Δ E2/ Δ E3, our thoughts are that as they still retain some protease activity (Fig 3C), this allows them to be internalised, albeit at possibly reduced rates. However, the 5 h incubation period that Ssp preparations were incubated with HEp-2 cells would have given the mutant Ssp sufficient time to enter the HEp-2 cells.

Yes, there was cell rounding observed for the Ssp- Δ E2 (~20%), Ssp- Δ E3 (~50%) and Ssp- Δ E2/ Δ E3 (~10%) mutants with Ssp- Δ L2 and Ssp-RAE identical to wildtype. The extent of cell rounding reflects their relative protease activities (Fig 3c). The difference in cell rounding of the different mutants is difficult to observe with the staining in Fig 5 and the 5 h incubation time needed to observe sufficient internalisation for figure images. Therefore, we added Fig 4a, without staining and a shorter incubation time, and approximated the % cell rounding given in the figure legend.

As per this reviewer suggestions and to further clarify these points we have added the following statements to the discussion:

(Page 8, line 324-327):

Ssp's subtilase activity was found to be critical for entry into human epithelial cells. Indeed, the Ssp-S341A active site mutant had no subtilase activity and was unable to enter Hep-2 cells, while the native as well as other proteolytically active Ssp variants, including Ssp- Δ E2 and Ssp- Δ E2/ Δ E3 which retained some protease activity, were internalised under the experimental conditions, albeit the latter protrusion mutants at possibly reduced rates.

(Page 8 lines 342-343).

Collectively, these results showed that the extent of cell rounding reflects the relative protease activity of the specific Ssp variant added (Fig. 4a and Fig. 5).

2: The relatively short β -helical stalk with the subtilisin domain directly on top is a unique AT structure not previously shown. One then wonders how well-studied subtilisin-like ATs (SphB1 and NalP) or other predicted subtilisin-like ATs would look. Do models of these passengers have the same unique features (protrusions around the active site, Ca²⁺ binding sites, short β -helix) and can this be modelled? This discussion is lacking; the authors choose only to compare the structure to SPATES and the esterase AT, for which structures are available, but that are clearly of a different AT class. However, the

comparison to other subtilisins would support the novelty and the understanding of the structural diversity of AT passengers in general.

As shown in FigS6 we did perform a sequence based alignment with other AT subtilases where we labelled putative active site residues, protrusions etc. With regard to modelling structures, overall Ssp shares on average 33% sequence identity to the other subtilase ATs, so we think modelling must be performed with some caution. We generated models of the NalP and SphB1 using AlphaFold and SWISS-MODEL, for which, models using the former program were more complete (Figure R1). Overall, the generated models for NalP and SphB1 show high similarity to the Ssp structure with both a subtilisin and β -helical stalk regions. Unfortunately, neither program was able to model the more unique E1-E3 active site protrusions well, with E1 being completely missing from both models, as the first 72 and 114 residues, were not modelled for NalP and SphB1, respectively. Given the poor quality of the models around the active site protrusions we decided not to add them to the paper but given them here for your inspection.

Figure R1. Shown is the crystal structure of **B.** Ssp (PDB: 8E7F) compared to the AlphaFold models of the **C.** NalP passenger in red (Genbank: AAN71715.1) and **D.** SphB1 passenger in green (Genbank: CAC44081.1) and with all structures overlaid **A.**

3. Line 73 and other places; remarks that “little information” is available on subtilisin ATs. In fact the functions of subtilisin-like ATs SphB1 and NalP have been studied in quite some detail, to an extent similar to most SPATEs. Importantly, both proteolytically process other virulence factors at the bacterial cell surface (filamentous haemagglutinin protein for *Bordetella*, IgA protease for *Neisseria meningitidis*). This is not mentioned anywhere in the manuscript, nor considered as a function for Ssp, whereas it is a striking similarity between subtilisins not shared with the SPATEs.

We have changed the statement to read “less is known” and we have added additional information for NalP and SphB1 with the corresponding references.

(Page 4, lines 74-77)

Some of the better studied subtilase ATs include *B. pertussis* SphB1 and *N. meningitidis* NalP, both lipidated^{17,18} and shown to cleavage different surface proteins such as filamentous hemagglutinin¹⁹ and the protease IgA or lactoferrin binding protein LbpB, respectively^{15,20}.

Function for Ssp: Given that SphB1 and NalP are the two most functionally characterised subtilase ATs we had compared Ssp to both of them and considered that Ssp could process other surface factors. However, both SphB1 (PMID: 12828647) and NalP (PMID: 23258267) are unusual in that they contain an N-terminal cysteine that is lipidated and keeps them tethered at the bacterial surface. This tethering at the bacterial surface has been associated with their abilities to process other bacterial surface factors. Given that Ssp lacks an N-terminal cysteine and is likely not lipidated we were cautious to state that Ssp remains at the bacterial surface to process other bacterial proteins.

other comments/suggestions:

4: The authors compare their structure in Fig S1 to the SPATE Pet. However, the cartoon shown appears not the most suitable: 1. the cartoon shown lacks secondary structure detail in its C-terminal part, most notably for the autochaperone domain and also the β -helical rungs that are present in Ssp. Better structures with detail in that region are available 2. the autochaperone domain of Ssp is comparable to that of IcsA, so one would like to see how that compares in this image as well.

As highlighted by this reviewer, the quality of the Pet crystal structure (4OM9) is limited (clearly shown by the wwPDB validation report), and this is reflected in the poor secondary structure assignment, particularly at the C-terminal end of the protein. However, with regard to autotransporters shown to enter epithelia, Pet is the protein of reference and, therefore, we consider important the structural comparison between Ssp and Pet. But following this reviewer recommendation and to improve the presentation of Pet, we have revised the figure and manually assigned the secondary structure based on structural comparison with other ATs of known structure (i.e. EspP, 3SZE) and the predicted secondary structure (PSIPRED). Additionally, with regard to providing a comparison to the C-terminal region of another AT, we have also included the structure of UpaB (6BEA). We thought UpaB might be the most suitable comparison as until Ssp it had the shortest β -helix to date, and like Ssp, includes the C-terminal autochaperone domain. Unfortunately, the structure of IcsA is not a complete structure of the passenger so it might not be a fair comparison. Nonetheless, our manuscript does include a comparison between the autochaperone domains of Ssp and IcsA (Fig S3).

5. The remark in line 57 (ATs “are increasingly gathering attention”) is overstating it. ATs are studied over 30 years and there is not a sudden increase in AT papers. This should be re-phrased.

The sentence has been rephrased to (Page 2, lines 56-59):

Autotransporters (ATs) are the largest family of secreted proteins in Gram-negative bacteria with many having important roles in bacterial infection and disease, including adhering to and invading host cells, biofilm formation along with being potent cytotoxins and immunomodulators^{6,7}.

REVIEWERS' COMMENTS

Reviewer #1 (Remarks to the Author):

This revised manuscript satisfactorily addresses the concerns that I discussed in my original review and that were identified by the other reviewer. I think the manuscript will be an important contribution to the literature, advancing our understanding of autotransporter proteins and subtilases.

Reviewer #2 (Remarks to the Author):

The manuscript "The first crystal structure of a subtilisin-like autotransporter passenger domain reveals insights into its cytotoxic function" by Hor et al. is an amended version of a manuscript that describes a novel crystal structure of an autotransporter passenger domain. The structure reveals a novel subtilisin sub-domain placed on top of a short β -helical stem, and first investigations on its putative function, based upon its functional subtilisin sub-domain and . They then investigate some of its functional aspects based upon structural information and the described functions for chymotrypsin-like autotransporters.

The authors have addressed the questions I raised in my original review in a satisfactory manner. The novelty of structure and the interesting functional aspects of it are well-described. The observed and interesting effects in the cell culture and Galleria assays were performed with either concentrated supernatants from E. coli cultures or purified proteins. This leaves the questions whether the amounts of Ssp used in these assays actually reflect levels expressed by Serratia itself during infections (this aspect is not really discussed). Furthermore, in view of the pleiotropic effects of other subtilisin-like and chymotrypsin-like autotransporters, other functionalities cannot be excluded. Nevertheless, the current work could be a good starting point to address such issues.

NCOMMS-22-40427B Revision

We thank the editor and all reviewers for their additional comments. Please see below our response to the suggested revisions.

REVIEWERS' COMMENTS

Reviewer #1 (Remarks to the Author):

This revised manuscript satisfactorily addresses the concerns that I discussed in my original review and that were identified by the other reviewer. I think the manuscript will be an important contribution to the literature, advancing our understanding of autotransporter proteins and subtilases.

We thank this reviewer for the very positive final comment

Reviewer #2 (Remarks to the Author):

The authors have addressed the questions I raised in my original review in a satisfactory manner. The novelty of structure and the interesting functional aspects of it are well-described. The observed and interesting effects in the cell culture and Galleria assays were performed with either concentrated supernatants from E. coli cultures or purified proteins.

We thank this reviewer for the very positive final comments.

- 1. This leaves the questions whether the amounts of Ssp used in these assays actually reflect levels expressed by Serratia itself during infections (this aspect is not really discussed). Furthermore, in view of the pleiotropic effects of other subtilisin-like and chymotrypsin-like autotransporters, other functionalities cannot be excluded. Nevertheless, the current work could be a good starting point to address such issues.*

To our knowledge there is not precise quantification of the amount of Ssp protein produced by the pathogen, however previous work has shown the presence of Ssp in the supernatant of nematocidal *Serratia* sp. A88copa13 at concentrations sufficient for the protein to be clearly detected by SDS-PAGE (PMID: 24244546). In this work, the Ssp concentration we used (25 µg/mL) was based on the concentrations previously used to characterise cytotoxic autotransporter proteases (i.e. 37 µg/ml for Pet autotransporter (PMID: 24327340) or 30 µg/ml for multiple SPATEs (PMID: 31198092)).

To address this point raised by the reviewer regarding adding detail about the concentration used, we have included the following statement in the methods Page 12, Lines 554-557 (shown in blue).

Confocal microscopy began with 25 µg/mL of concentrated supernatants of Ssp variants added per well and the cells incubated for 5 h before washing with PBS (this concentration is based on the concentrations previously used for the characterisation of cytotoxic autotransporters (30-37 µg/mL)^{24,67} as well as the Ssp supernatant concentration previously detected by SDS-PAGE in nematocidal *Serratia* sp.²⁸).